METHODS

# A graph neural network-based spatial multi-omics data integration method for deciphering spatial domains

Congqiang Gao[1☯], Chenghui Yang[2,3☯], Lihua Zhang [2,3]*

**1** School of Cyber Science and Engineering, Wuhan University, **2** School of Computer Science, Wuhan University, **3** School of Artificial Intelligence, Wuhan University, Wuhan, China

☯ These authors contributed equally to this work.
* zhanglh@whu.edu.cn

## Abstract

Recent advancements of spatial sequencing technologies enable measurements of transcriptomic and epigenomic profiles within the same tissue slice, providing an unprecedented opportunity to understand cellular microenvironments. However, effective approaches for the integrative analysis of such spatial multi-omics data are lacking. Here, we propose SpaMI, a graph neural network-based model which extract features by contrastive learning strategy for each omics and integrate different omics by an attention mechanism to integrate spatial multi-omics data. We applied SpaMI to both simulated data and three real spatial multi-omics datasets derived from the same tissue slices, including spatial epigenome–transcriptome and transcriptome–proteome data. By comparing SpaMI with the state-of-the-art methods on simulation and real datasets, we demonstrate the superior performance of SpaMI in identifying spatial domain and data denoising.

### Author summary

The rapid advancement of spatial sequencing technologies now enables the simultaneous measurement of multiple omic features within the same spot. However, high levels of data noise and inherent sparsity pose significant challenges to the effective integration of such spatial multi-omics data. To address this, we present SpaMI, a deep learning-based model designed to integrate spatial multi-omics data and produce a complementary, comprehensive representation. SpaMI incorporates a contrastive learning strategy, an attention mechanism, and cosine similarity regularization. The contrastive learning component facilitates more effective learning of modality-specific embeddings while mitigating the influence of noise, and the attention mechanism adaptively aggregates embeddings across different modalities. The resulting cell representation offers powerful support for downstream analyses such as cell clustering, spatial domain identification, and differential expression detection.

**Data availability statement:** An open-source Python implementation of the SpaMI toolkit is accessible at https://github.com/Gaocongqiang/SpaMI.

**Funding:** This work has been supported by the National Key Research and Development Program of China (no. 2023YFF0725400 to L.Z.), National Natural Science Foundation of China (no. 62202343 to L.Z.) and the Fundamental Research Funds for the Central Universities (2042024kf0027 to L.Z.). The funders had no role in study design, data collection and analysis, decision to publish, or preparation of the manuscript.

**Competing interests:** The authors have declared that no competing interests exist.

## Introduction

Single-cell sequencing technologies have emerged as a pivotal advancement in biological sciences this century, enabling researchers to explore gene expression, chromosome accessibility, and other biomolecular characteristics at the individual cell level [1,2]. However, spatial location information of cells is lost by single cell sequencing technologies. The advent of spatial transcriptomics technologies has furnished robust tools for exploring the spatial distribution of cells [3,4]. Recently, spatial sequencing technologies have progressed from measuring single-omics on individual sections to conducting multi-omics sequencing on the same tissue section [5], which allows for more comprehensive analysis of cellular microenvironments [6]. Technologies such as DBiT-seq [7], SPOTS [8], spatial CITE-seq [9], SM-Omics [10], and Stereo-CITE-seq [11] can simultaneously sequence the spatial transcriptome and spatial proteome [12]. Alternatively, methods like MISAR-seq [13], spatial ATAC-RNA-seq [14], and spatial CUT&Tag-RNA-seq [14] target the spatial transcriptome and spatial epigenome [15] simultaneously. MiP-seq [16] can simultaneously detect DNA, RNA, proteins, and biomacromolecules at a subcellular resolution. These technologies offer a comprehensive and complementary view for understanding of cellular and tissue properties [17].

Some methods have been proposed for integrating single cell multi-omics data. For example, Seurat V4 [18] is based on the weighted nearest neighbor principle, mapping multi-omics feature data into a unified low-dimensional space to calculate cell distances, and generating integrated feature vectors by weighted average of adjacent cell features. MOFA+ [19] relies on the Bayesian group factor analysis framework, which models multi-omics data through shared latent variable factors. In addition, methods such as MultiVI [20], totalVI [21], and scMM [22] also achieve joint characterization and information fusion of single-cell multi-omics data through different method designs. However, spatial multi-omics data are inherently heterogenous and highly sparse, which poses significant challenges. It is inappropriate to directly migrate these methods to the integration of spatial multi-omics data. They do not take into account the use of spatial information at all, which is of great help in analyzing the spatial dependencies and interactions between cells.

Recently, SpatialGlue [23] has been proposed to integrate spatial multi-omics data within the same tissue section. It employs attention mechanisms at various levels to effectively fuse feature maps and spatial representations across different modalities. The model adopts single-layer GCN as encoder, making the complex spatial features might not be fully captured [24,25]. Li et al. proposed the PRESENT framework [27], which was grounded in a multi-view autoencoder architecture. The framework employed graph attention networks and Bayesian networks to enable spatial multi-omics data integration. Tian et al. proposed the spaMultiVAE [26] model, an extension of spaVAE [26], which utilizes gaussian process to regularize omics patterns with spatial coordinates and it usually needs large computation complexity. An efficient spatial multi-omics data integration method is still needed to reveal heterogeneously spatial structures and investigate the underlying biological functions from multi-views [28].

To bridge the gap between the burgeoning spatial multi-omics sequencing technologies and the limited methods designed for integrating such data, we introduced SpaMI—an efficient, universally applicable deep learning approach designed for cross-modal representation of spatially-aware multimodal data from the same tissue section. SpaMI uses a graph convolutional network(GCN) [24] to encode the spatial neighbor graph based on the shared spatial positions and the expression profiles specific to each omics data, and uses a contrastive learning strategy to refine the low-dimensional embedding of each modality. With these embeddings, SpaMI adopts an attention aggregation mechanism to adaptively learn the importance of different modalities, allowing for more accurate integration which could be used for subsequent downstream analyses, such as low-dimensional visualization, detailed spatial domains identification and data denoising. We first validated SpaMI on multiple simulated datasets, where it demonstrated excellent and robust performance through spatial visualization and quantitative comparisons. Applications to mouse embryonic brain and juvenile mouse brain datasets confirmed SpaMI's efficacy in integrating spatial transcriptomes and epigenomes, identifying precise spatial structures with higher resolution, and denoising the dataset. Further experiments on mouse thymus dataset, integrating spatial transcriptomes and proteomes, showed that SpaMI could effectively utilize information from various omics in a complementary manner, allowing for a more comprehensive resolution of spatial structures. These results demonstrate that SpaMI is a cross-platform, robust spatial multi-omics integration tool that can achieve more refined identification of spatial domains. Totally, SpaMI is able to provide multimodal and complementary cellular perspectives when integrating spatial multi-omics data while maintaining shared and modality-specific cell biological variations. These features provide great potential for the systematic and comprehensive exploration of gene regulatory processes and cellular activities in the context of tissue microenvironment, such as exploring cancer cell subpopulations with different metastatic potentials in tumor tissues and cell differentiation processes in tissue spatial structures.

## Result

### Overview of SpaMI

SpaMI is a deep learning model utilizing graph autoencoders, which integrates spatial multi-omics data sequenced from the same tissue slice to decipher spatial domains of tissue samples from a comprehensive and high-resolution perspective (Fig 1). SpaMI first builds the graph with each spot as one node and the edges are connected based on the spatial coordinates (Methods). As the data comes from the same slice, the graphs for different omics are the same. Although the graphs of different omics have identical topological structure, the characteristics of the nodes are different. We adopt a contrastive learning strategy, similar to deep graph infomax (DGI) [29] to capture the spatially dependent patterns. Specially, we build a corrupted graph by randomly shuffling the features while keeping the original graph's topological structure unchanged. Then SpaMI obtains low-dimensional embeddings of each omics data on spatial graph and corrupted graph through two-layers graph convolutional encoders. Next, SpaMI maximizes the mutual information between the low-dimensional embeddings of the spot and the local context (Methods). With the graph encoders, SpaMI obtains omics-specific latent representations $Z_1$ and $Z_2$. A cosine similarity regularization is incorporated to restraint $Z_1$ and $Z_2$. SpaMI obtains an integrated embedding $Z$ with an attention mechanism on $Z_1$ and $Z_2$. After that, the integrated embedding $Z$ is inputted back into the omics-specific decoders for reconstructing data of each omics. Finally, the integrated embedding and the reconstructed data are used for downstream analyses such as spatial domain identification, spatial variable genes' detection and revealing spatial specific transcriptional regulatory links.

### Comparison of SpaMI with competing methods on simulated spatial multi-omics datasets

We generated simulated spatial multi-omics data to test the integrative ability of SpaMI. Specially, the simulated data were generated based on the methodology described in the previous study [30]. For the transcriptome, we simulated gene expression matrices using a zero-inflated negative binomial (ZINB) distribution [31]. Proteome expression profiles were simulated using a negative binomial (NB) distribution [23], and chromatin accessibility matrices were modeled with a zero-inflated

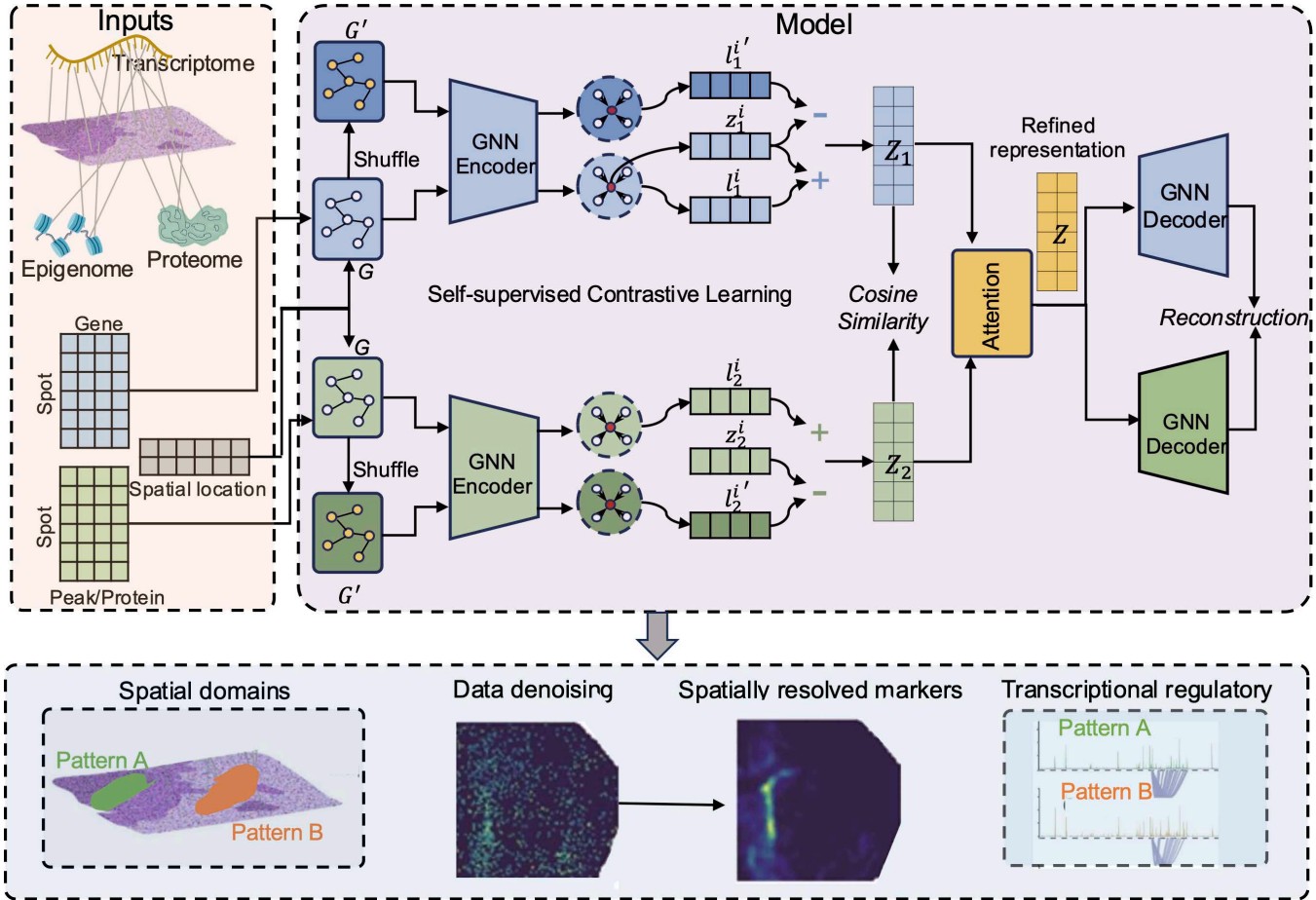

**Fig 1. Overview of SpaMI.** The inputs of SpaMI are spatial multi-omics data including transcriptomic profile, epigenomic profile or proteomic profile and spatial location. SpaMI builds a spatial neighbor graph $G$ based on spatial location. The corrupted graph $G'$ is built by randomly shuffling features of each node. The GNN encoder is applied to both spatial graph and the corrupted graph for each omics. $z_j^i$ is the local context vector of the $i$-th spot from the $j$-th omics computed by averaging the low dimensional representations of its neighbors. $l_j^i$ and $l_j^{i'}$ are the local representations of the $i$-th spot from the $j$-th omics of spatial graph and corrupted graph. $z_j^i$ and $l_j^i$ are a positive sample pair, while $z_j^i$ and $l_j^{i'}$ are a negative sample pair. By the contrastive learning, SpaMI obtains the omics-specific representation $Z_1$ and $Z_2$. The relationship between $Z_1$ and $Z_2$ is regularized by a cosine similarity constraint. Then $Z_1$ and $Z_2$ are adaptively integrated through the attention aggregation mechanism to obtain the refined spot latent representation $Z$, which is then inverted back to the original feature space of the different modalities by GNN decoder. Then the latent representation $Z$ and reconstructed data are used for downstream analyses, such as identifying spatial domains, denoising spatial data, identifying spatially resolved markers and detecting spatial specific transcriptional regulatory links.

Poisson (ZIP) distribution [32]. We use the NSF [30] model to extract spatial factors on a 2D spatial grid, capture the similarity in feature expression between neighboring spatial locations, and reconstruct a feature expression matrix with spatial dependencies to simulate the data. We also introduced Gaussian noise to these simulations to mimic real-world data conditions. Each data comprises four distinct factors and backgrounds, collectively representing five cell types. We divided these data into two scenarios including one dataset contains spatial transcriptomic and spatial proteomic data, and another dataset contains spatial transcriptomic and spatial epigenomic data (Table A in S1 Text). We benchmarked SpaMI against Seurat, MOFA+ and SpatialGLUE using four metrics including adjusted Rand index (ARI), adjusted mutual information (AMI), normalized mutual information (NMI), homogeneity score (Homo) to quantitatively evaluate the integrative performance of these methods. Specially, we identified clusters on the integrated space of these methods by leiden algorithm [33].

SpaMI accurately identified four factors as well as their backgrounds on both integrating transcriptomics and pro-teomics data, and integrating transcriptomics and chromatin accessibility data (Fig 2A). SpatialGlue detected the factors with some noise. Moreover, on the second simulated RNA+ATAC data, SpatialGLUE failed to detect the factor edges, while spaMI accurately detected the factors and background (Fig A in S1 Text). Whereas the other two methods could only partially recover the factors and failed to detect the factor 3. Different factors were more dispersed in the integrated space of SpaMI than other methods (Fig 2B). To test the robustness of integrative methods, we added noise to the simulated data with increasing levels of Gaussian noise. As expected, the performance of most methods decreases with increasing levels of Gaussian noise. SpaMI consistently outperformed the other methods with highest metrics (Fig 2C). Spatial methods SpaMI and SpatialGlue performed better than single cell multi-omics data integrative methods Seurat and MOFA+, indicating that spatial information contributes greatly to spatial multi-omics integration. SpaMI was consis-tently demonstrated superior performance than other methods in integrating spatial transcriptomic and chromatin acces-sibility data (Fig A in S1 Text). Next, we conducted an ablation study to check the contributions of specific components within the SpaMI model by individually removing the cosine similarity loss, contrastive learning loss, and substituting the attention aggregation mechanism with a simple concatenation operation. We observed that all these components played an important role in integrating spatial multi-omics data and removing any component would decrease the model's per-formance (Fig 2D and Fig I in S1 Text).

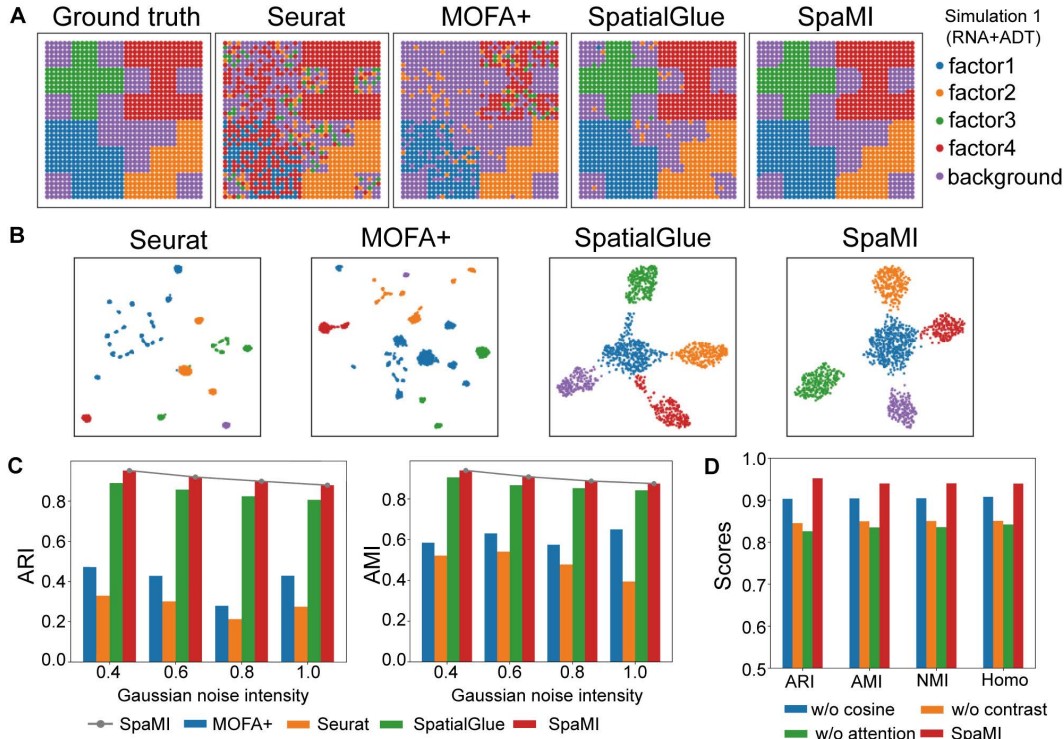

**Fig 2. Comparison of SpaMI with other methods on simulated spatial multi-omics datasets. (A)** Spatial plots of the simulated data (RNA+ADT), from left to right: Ground truth, Seurat, MOFA+, SpatialGlue, SpaMI. **(B)** UMAP visualization of Seurat, MOFA+, SpatialGlue and SpaMI on simulated data (RNA+ADT). **(C)** Quantitative evaluation of the four methods using ARI, AMI, NMI and Homo metrics. The horizontal axis represents the gradually increasing standard deviation of Gaussian noise. **(D)** Results of ablation experiment of SpaMI model on the simulated RNA+ADT modality dataset.

## SpaMI demonstrates superior performance in characterizing the structure of mouse embryonic brains

We applied SpaMI to the spatially resolved chromatin accessibility and gene expression data of moue embryonic brain at E15.5 stage coming from MISAR-seq [13] platform. We performed manual annotation (Fig 3A) of the dataset by referencing the anatomical annotation structure in the previous study [13], and used this to evaluate the clusters detected by Leiden algorithm [33] on the SpaMI's embedding. We compared SpaMI against Seurat, MOFA+ and SpatialGLUE using ARI, AMI, NMI and Homo metrics. SpaMI had highest values in all four metrics than other methods (Fig 3B and Fig B in S1 Text). In addition, we employed the Squidpy [34] package to compute the Moran's I score, which measures the spatial distribution pattern of clustered cell types. Higher scores indicate more pronounced spatial clustering [35], where SpaMI and SpatialGlue emerged as the top performers, confirming the efficacy of SpaMI in spatial clustering visualization (Fig 3C). SpaMI excels at identifying distinct regions within the mouse embryonic brain, aligning closely with manual annotations (Fig 3D). For example, SpaMI accurately distinguished Dpallm and Dpallv, and provided precise identification of the spatial structures of the Midbrain, Muscle, and Skull areas. In contrast, other methods struggled with these tasks. Specially, SpatialGlue was imprecise in capturing the Skull region, MOFA+ had difficulties in differentiating between the Midbrain and Cerebellar vermis, and Seurat incorrectly identified regions in the Midbrain and Skull.

We subsequently investigated the denoising capability of SpaMI. We converted the reconstructed data to gene expression profiles by SpaMI and compared it with the original normalized gene expression. The reconstructed gene expression was more localized spatially, with significantly reduced noise (Fig 3E). Specially, the denoised ones by SpaMI exhibited the structure-marker genes clearly. For example, after denoising, the muscle specific marker gene Myh3 showed strong expression signals in Muscle region, while its original spatial expression was messy. To further assess SpaMI's proficiency in integrating two modalities in a complementary manner, we conducted an experiment where we removed two key model components (the attention aggregation mechanism and cosine similarity loss) and restricted the model to process only one modality at a time. This alteration revealed that the RNA modality data was particularly effective in identifying Muscle regions, while the ATAC modality was crucial for distinguishing between the Dpallm and Dpallv regions (Fig 3F). Importantly, SpaMI demonstrated a unique capability to capture and integrate the heterogeneity present in both modalities, achieving a more comprehensive and accurate identification of various regions by integrating RNA and ATAC data.

## SpaMI identifies more fine-grained structures in coronal slices of juvenile mouse brain

Next, we applied SpaMI to a spatial epigenome-transcriptome data obtained using spatial ATAC-RNA-seq [14] technology on coronal sections of 22-day-old mouse brains. We utilized the Allen brain atlases [36] to annotate clusters identified by Leiden algorithm on embedding of SpaMI. And we used the manual annotation (Fig 4A) of tissue from P22 mouse brain coronal sections in Zhang et al. [37] as ground truth. We compared SpaMI with Seurat, MOFA+ and SpatialGLUE (Fig 4A-4D and Fig C in S1 Text). Seurat and MOFA+ were two non-spatial methods, which roughly follow the expected layer pattern in this section, but the boundary of its clusters was discontinuous with many outliers, which impaired its clustering accuracy analyses from Seurat and MOFA+ exhibited considerable noise within the original data, which significantly impeded the identification of cellular structures. Seurat was only able to delineate a few regions such as the lateral preoptic (lpo), lateral ventricle (vl), and caudoputamen (cp), while MOFA+ identified even fewer regions. In contrast, SpaMI substantially achieved clear delineation of various regions, aligning closely with annotations in the Allen brain atlas. Notably, SpaMI precisely identified complex and smaller regions like 3-ccg/aco, 5-acb/11-ccb, 16-vl, and 18-lpo, and provided a more detailed segmentation of cortical layers. Although SpatialGlue also performed well in accurately partitioning spatial regions of the mouse brain, SpaMI delivered results with less noise and more defined boundaries between spatial regions. A particular strength of SpaMI was evident in its handling of the nucleus accumbens region (acb). While SpatialGlue inaccurately grouped a significant portion of this region with the caudoputamen (cp), SpaMI distinctly divided it into 5-acb and 11-acb, achieving finer discrimination from the 1-cp region. This superior delineation by SpaMI is further supported by the Moran's I score and four clustering metrics (ARI, AMI, NMI and Hono), indicating a

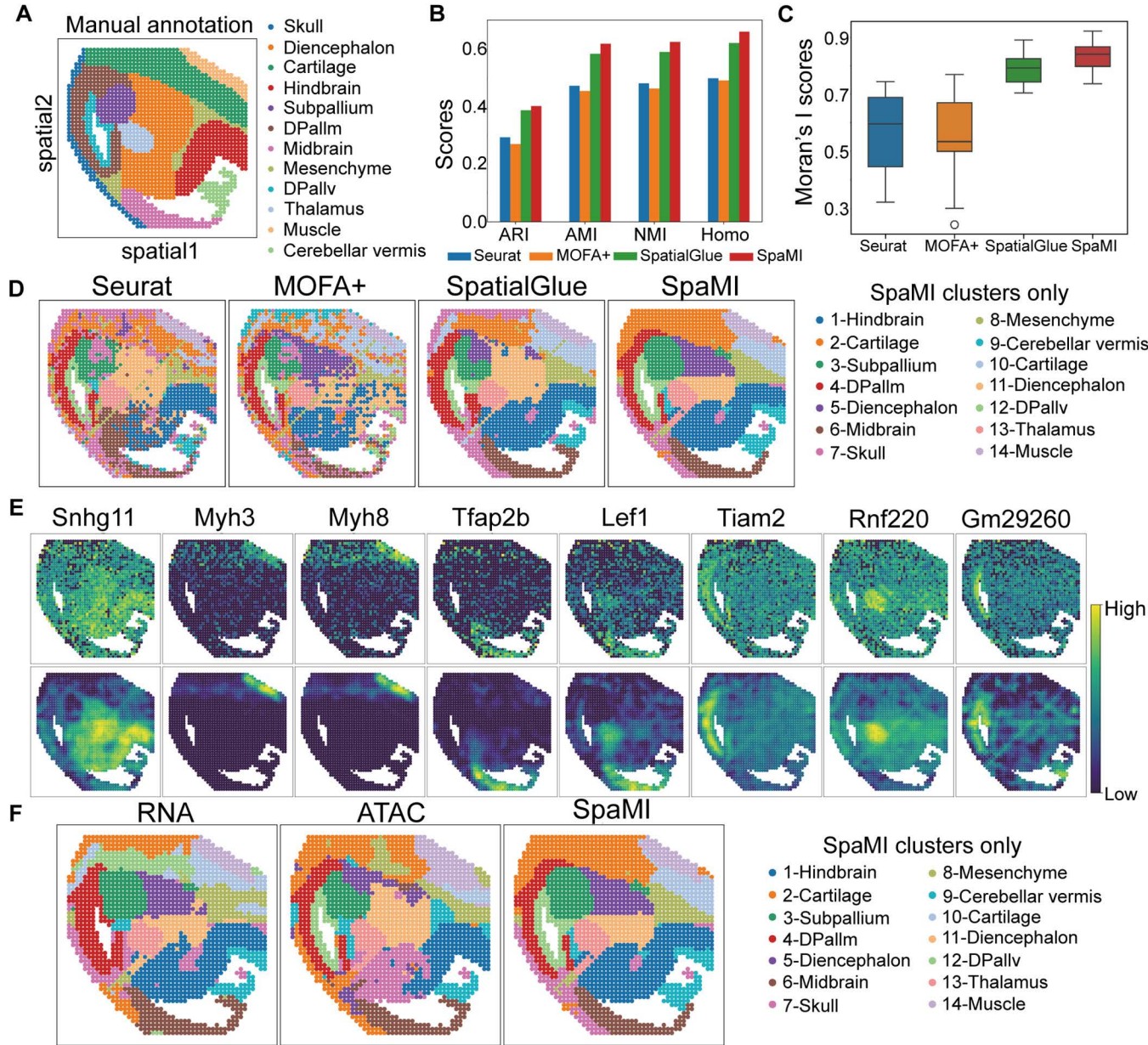

**Fig 3. SpaMI demonstrates superior performance in characterizing the structure of mouse embryonic brains. (A)** Manual annotation of mouse embryonic brain. **(B)** Quantitative evaluation of the four methods using ARI, AMI, NMI and homogeneity metrics. **(C)** Box plots of Moran's I scores of Seurat, MOFA+, SpatialGLUE and SpaMI. **(D)** Spatial plots of the mouse embryonic brains data, from left to right: Seurat, MOFA+, SpatialGlue, SpaMI. **(E)** Visualizations of the original normalized spatial expression of marker genes (top) and denoised ones obtained by SpaMI. **(F)** Visualization of spatial clustering results for model using only single modality data and combining both modality data, from left to right, RNA modality data alone, ATAC modality data alone, and both modality data combined.

strong spatial correlation among cells of the same type and high accuracy in identifying different cell types, with SpaMI leading the results followed by SpatialGlue (Fig 4B-4C). We also conducted a visual comparison between the reconstructed and the original gene expression profiles (Fig 4E). Following denoising by the model, the expression of marker genes displayed consistent spatial patterns with specific cell types, leading to markedly clearer distinctions between

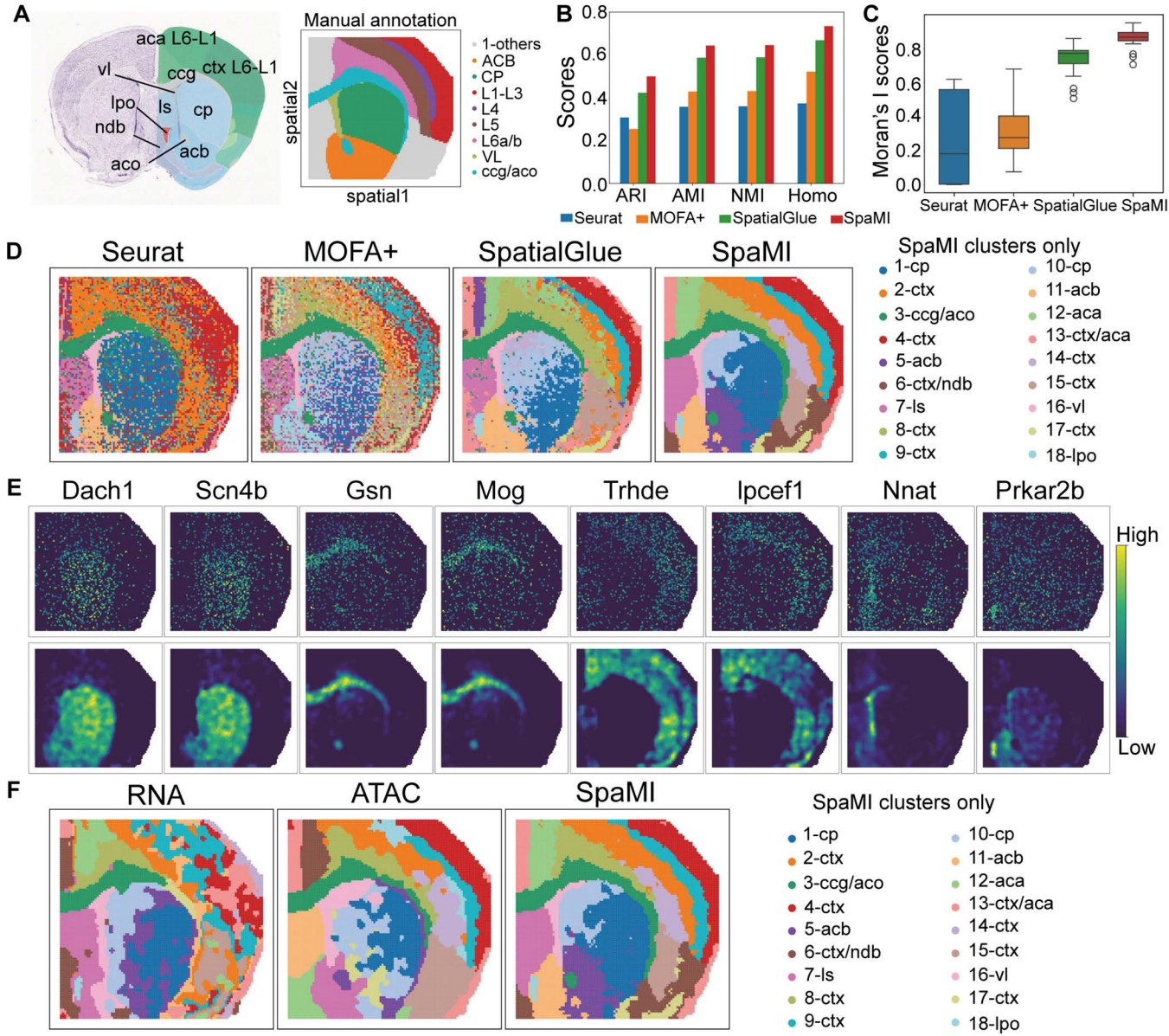

**Fig 4. SpaMI identifies more fine-grained structures in coronal slices of juvenile mouse brain. (A)** Annotated reference of the mouse brain coronal section from the Allen Mouse Brain Atlas (left) and Manual annotation of mouse brain (right). **(B)** Quantitative evaluation of the four methods using ARI, AMI, NMI and homogeneity metrics. **(C)** Box plots of Moran's I score of the four methods. **(D)** Spatial plots of the mouse brain data, from left to right: Seurat, MOFA+, SpatialGlue, SpaMI. **(E)** The original spatial expression (top) and reconstructed spatial expression (bottom) of some marker genes corresponding to the spatial domain identified by SpaMI. **(F)** Visualization of spatial clustering results for model using only single modality data and combining both modality data, from left to right, RNA modality data alone, ATAC modality data alone, and both modality data combined.

different spatial regions. Additionally, we experimented with inputting only single-modality data into a modified version of our model (Fig 4F). RNA modality played a pivotal role in identifying the overall contours of non-cortical regions, such as the caudoputamen (cp). It reliably pinpointed the lateral preoptic area (lpo) as well. Conversely, the ATAC modality was instrumental in distinguishing various cortical layers and was particularly effective in identifying structures within the anterior commissure regions (aco).

SpaMI can be extended to SpaMI-P (Fig E in S1 Text) to leverage the prior data coming from public databases such as Mouse Allen Brain Atlas like PAST [44]. Specially, SpaMI-P model incorporates a prior information encoder and MNN [17] alignment strategy to the framework of SpaMI, enabling it to utilize labeled prior modal data from similar tissue sections for training guidance. We found that SpaMI-P well divided the cortical layer than SpaMI. And SpaMI-P had higher Moran's I score than that of SpaMI (Fig E in S1 Text). These experimental results well illustrate that prior information play an important role in guiding the model training process.

Next, we applied an optimal transport-based method FGOT (https://github.com/lhzhanglabtools/FGOT) to infer the transcriptional regulatory links based on the embedding obtained by SpaMI. Previous study has illustrated that PcP4 is broadly expressed throughout the cortex [38]. We found that two specific regulatory links highlighted with red color were specific to the 4-ctx cluster, which was close to L2/3 based on the annotation. Interestingly, there were strong H3K4me1 signals around these two links, which indicated that these two regulatory elements were functional enhancers (Fig 5A). To systemically investigate whether the cortex clusters capture the functional cis-regulatory elements, we performed differently expressed gene detection by Wilcox rank test and identified markers with adjust p-value being less than 0.05 and log-foldchange being higher than 0.5. Then we computed the average H3K4me1 signals of L2/3, L4 and L5 layers about 1000 bp upstream and 1000 bp downstream from the middle base of functional peaks, which detected by FGOT on the embedding of SpaMI. We also computed the average H3K4me1 signals of the rest regions using bigWigAverageOverBed. We found that the functional peaks had significantly strong H3K4me1 signals than that of rest regions (Fig 5B). Moreover, the functional peaks of 9-ctx exhibited higher H3K4me1 signals of L4 than that of L2/3 and L5 layers, indicating that the cluster 9-ctx was highly enriched by L4 layers. Totally, SpaMI is an efficient spatial multi-omics integration tool, which provides a meaningful embedding for FGOT to infer transcriptomic regulatory links.

## SpaMI accurately deciphers the spatial structure of thymus through spatial clustering

We applied SpaMI to mouse thymus protein and mRNA data obtained using Stereo-CITE-seq [11] technology, broadening our validation to demonstrate that SpaMI is highly adaptable across multiple technological platforms and modality combinations. The mouse thymus, as characterized in existing studies [39–41], features a bilobed structure with each lobe divided into two principal regions: the cortex and the medulla. The cortex is enriched in the outer layer, primarily consists of immature T cells, epithelial cells, dendritic cells, and macrophages. While the medulla is enriched in the inner layer, harbors mature T cells and distinct structures such as thymic corpuscles. Both cortex and medulla are thought to play a crucial role in regulating immune tolerance. This complex organization of the thymus creates an efficient environment for the proper development of T cells and the maintenance of immune system functionality [42,43].

The spatial clustering results on the embedding of SpaMI were highly consistent with the bilobed structure of the mouse thymus (Fig 6A and Fig F in S1 Text). Specifically, the 7,9-Connective tissue capsule delineates the thymus from surrounding tissues, while the 8-Connective tissue capsule separates the two lobes. Each lobe is distinctly composed of a medulla surrounded by various cortical layers(2,3,1-Cortex) with a transition area, the 6-Corticomedullary junction, between the cortex and medulla. These observations are consistent with known studies [39–41]. While SpatialGlue captured the general structure of the thymus and identified two cortical layers, SpaMI provided a more granular subdivision, distinguishing three distinct layers: inner, middle and outer, thereby revealing a more detailed and nuanced structure (Fig 6A). Conversely, MOFA+ struggled with finer division of the cortex, and Seurat was unable to accurately identify the cortex and corticomedullary junction, capturing the least structural detail. In terms of Moran's I score, both SpaMI and Spatial-Glue performed better than Seurat and MOFA+ (Fig 6B-C). Furthermore, we assessed SpaMI's denoising capabilities and observed a significant reduction in noise within the reconstructed spatial gene expression compared to the original data (Fig 6D). Our analysis revealed differential expression of the marker genes Ccr7, Klf2, Rag1, and Trbc1 in the cortex (clusters 1, 2, and 3) and medulla (cluster 5), which correlates with the distribution of T cells at different developmental stages (Fig 6E). Double-positive T cells and double-negative T cells are predominantly present in the cortical region,

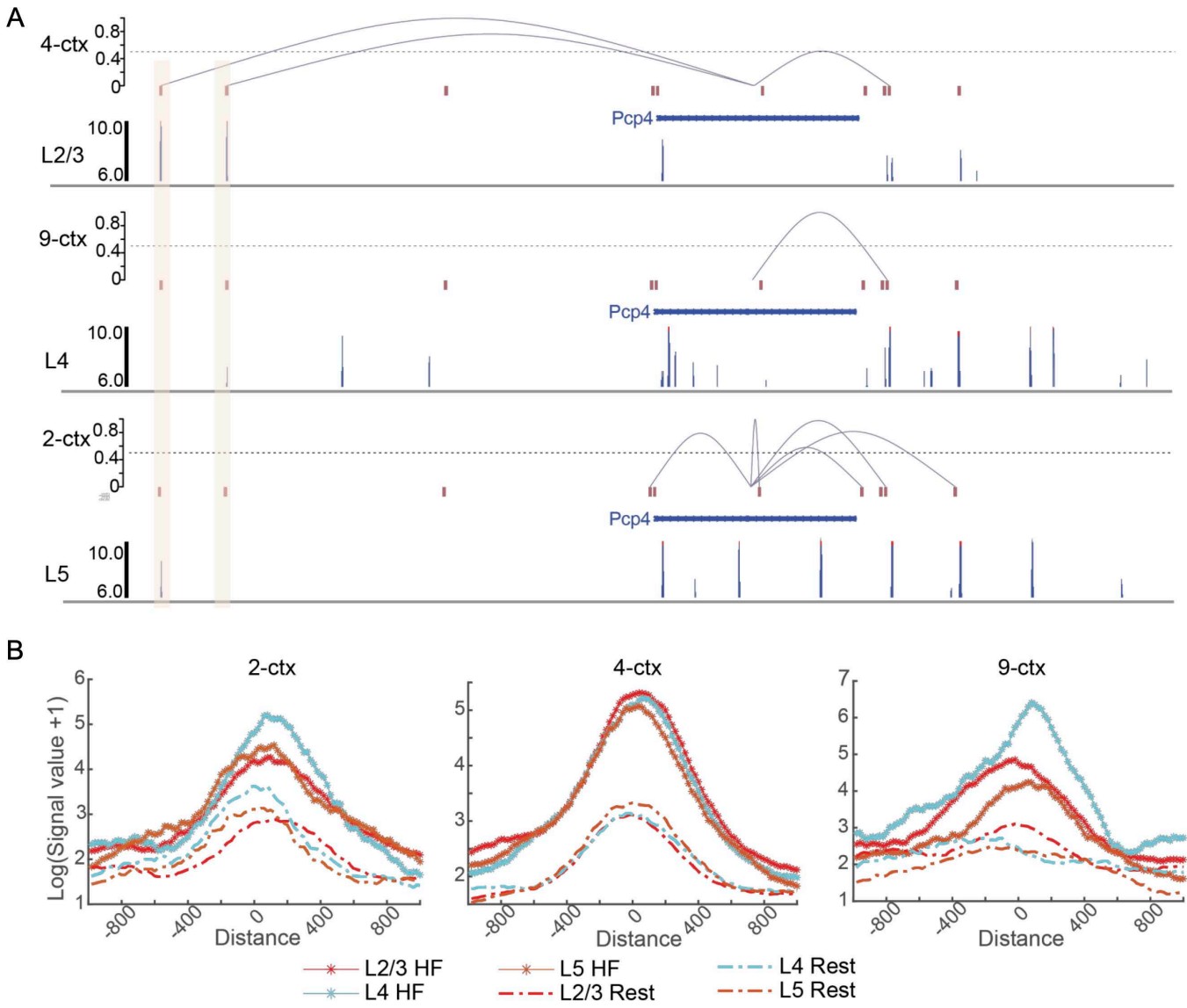

**Fig 5. The spatial specific regulatory links inferred on the embedding of SpaMI in cortex layers. (A)** Regulatory links for the Pcp4 locus with H3K4me1 signal of L2/3, L4 and L5. All the regulatory scores are normalized to 0-1. Two regulatory links are highlighted in a light red color. **(B)** Average H3K4me1 signals of L2/3, L4 and L5 layers about 1000 bp upstream and 1000 bp downstream from the middle base of functional peaks and rest regions for three cerebral cortex clusters using bigWigAverageOverBed. Each value is computed by averaging over 20-bp bin.

where they express the Ccr7 and Klf2 genes while still immature. In contrast, single-positive T cells are predominantly present in the thymic medulla, where they express the Rag1 and Trbc1 genes, and these cells are close to maturity, ready to transition to the peripheral immune system. Additionally, we noted that differential expression of the CD169 marker significantly aided the model in accurately identifying the 8-Connective tissue capsule (Fig 6E). This observation aligns with our further experiments utilizing solely ADT modality data. Although ADT data effectively pinpointed the connective tissue capsule, it introduced more noise. Conversely, RNA data proved more effective in delineating the cortex and medulla with reduced noise (Fig 6F). The synergistic integration of these modalities not only clarifies but also enriches our understanding of the spatial architecture of the mouse thymus, revealing a more detailed and comprehensive structural depiction.

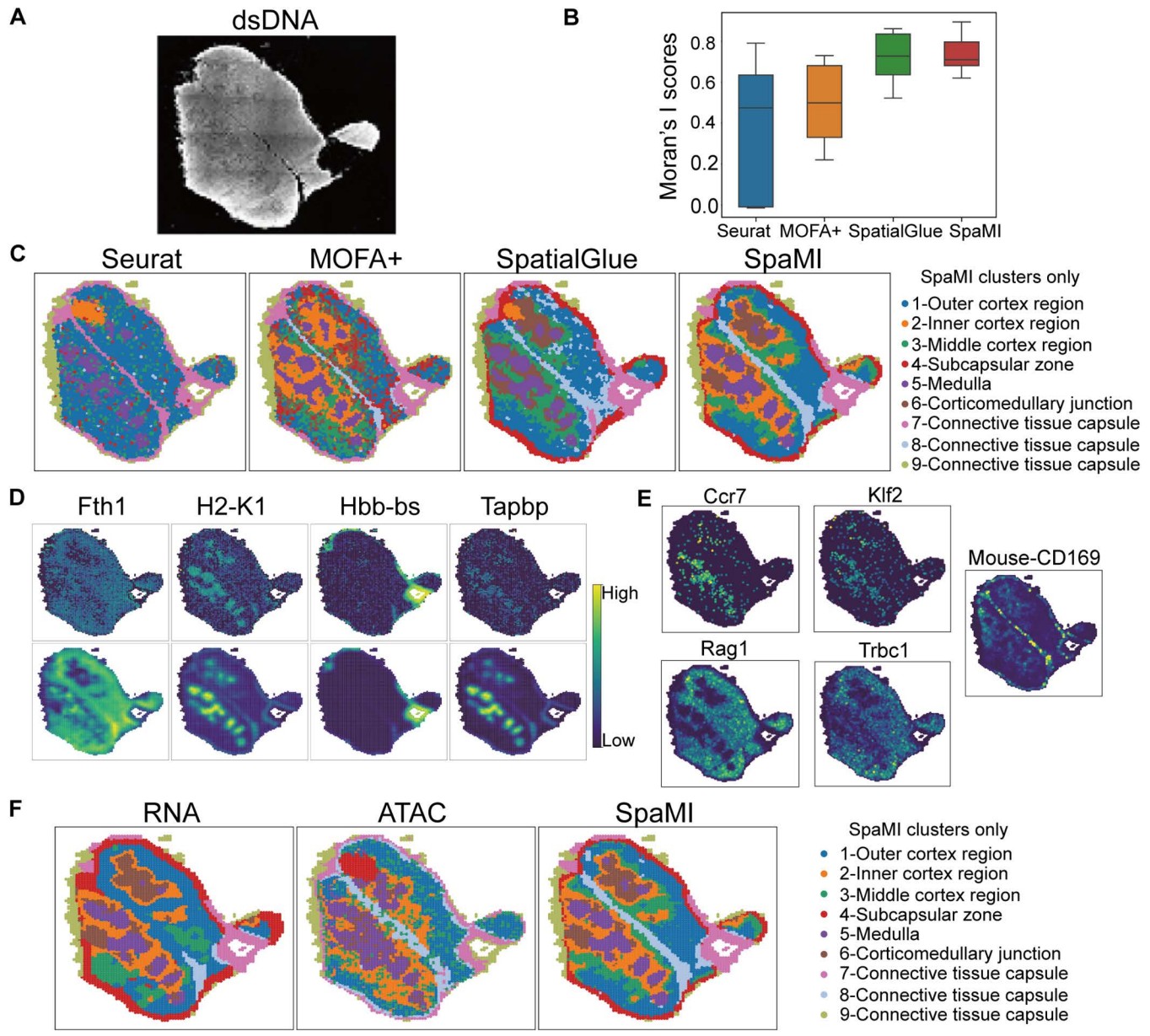

**Fig 6. SpaMI more accurately deciphers the spatial structure of mouse thymus through spatial clustering. (A)** dsDNA image of mouse thymus. **(B)** Box plots of Moran's I score of the four methods. **(C)** Spatial visualization of Seurat, MOFA+, SpatialGlue and SpaMI on the mouse thymus data. **(D)** The original spatial expression (top) and reconstructed spatial expression (bottom) of some marker genes corresponding to the spatial domain identified by SpaMI. **(E)** Spatial visualization of the expression levels of marker genes and proteins. **(F)** Visualization of spatial clustering results for model using only single modality data and combining both modality data, from left to right, RNA modality data alone, ATAC modality data alone, and both modality data combined.

We also applied SpaMI to the human lymph node dataset generated by 10x Genomics Visium RNA and protein co-profiling technology. We leveraged the manual annotations by Long et al. [23] for evaluation (Fig 7A). The anatomical structure of this biological tissue includes the cortex and medulla in the core region, as well as the surrounding pericapsular adipose tissue and capsule. SpaMI demonstrated the highest correspondence to annotated cortical regions (Fig 7B-E).

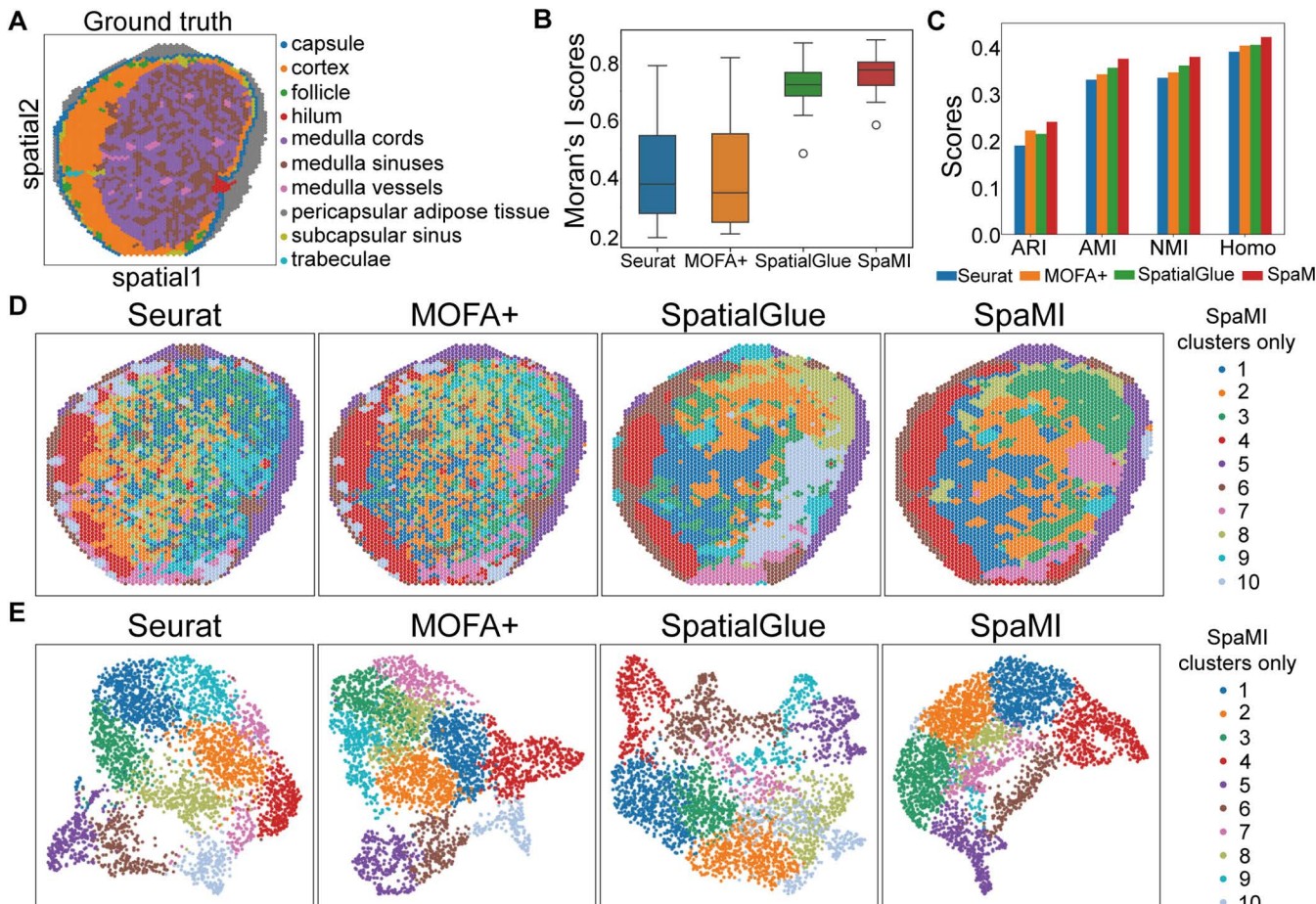

**Fig 7. The performance of SpaMI on human lymph node dataset. (A)** Manual annotation of human lymph node sample. **(B)** Box plots of Moran's I score of the four methods. **(C)** Quantitative evaluation of the four methods using ARI, AMI, NMI and homogeneity metrics. **(D)** Spatial plots of the human lymph node data, from left to right: Seurat, MOFA+, SpatialGlue, SpaMI. **(E)** UMAP visualization of Seurat, MOFA+, SpatialGlue and SpaMI on human lymph node data.

SpatialGlue and Seurat failed to identify partial adipose tissue. For quantitative assessment, we used the unsupervised Moran's I score to evaluate spatial autocorrelation of cell clusters, where SpaMI and SpatialGlue achieved high scores (Fig 7B). Supervised evaluation based on the manual annotations further showed that SpaMI outperformed all other methods in four metrics. These results validate SpaMI's effectiveness on both mouse and human samples, and highlight its utility in spatial multi-omics integration.

## Discussion

The advent of spatial omics technology has revolutionized our ability to capture the spatial distribution of multiple omics features within the same tissue section. While different omics datasets often share information that can elucidate tissue structure, they also contain unique elements that clarify multi-level tissue relationships. Thus, transitioning from single omics analysis to a combined approach that integrates spatial data with multiple omics layers offers a holistic and complementary perspective, essential for delineating continuous spatial tissue domains with biological significance. However, the integration of disparate omics data, each with its own distinct characteristics, poses significant challenges. The utilization

of spatial information also affects the integration effect. To address these challenges, we developed SpaMI, a novel graph neural network model designed to synthesize diverse omics perspectives and seamlessly integrate them. This integration is achieved through an attention aggregation mechanism that adaptively adjusts the weights of low-dimensional embeddings from different modalities. This capability enables SpaMI to effectively highlight crucial features across multi-omics data. Moreover, SpaMI employs a contrast learning strategy that not only associates the spot representations of each modality with their local neighborhood contexts but also minimizes the influence of noise on these representations.

Although both SpatialGlue and spaMI adopted graph neural networks (GNN) and attention mechanisms, there are major differences between spaMI and SpatialGLUE. There are three parts in SpaMI model, which are contrastive learning strategy, attention mechanism and cosine similarity regularization. We have conducted ablation studies to validate that each part improves the integration performance. The contrastive learning strategy was added to the graph neural network to extract spatially informative features. In addition, we added cosine similarity regularization to better extract the homogeneity between different omics. Ultimately, SpaMI provides a robust framework for exploring the complex interplay of spatial multi-omics data, paving the way for groundbreaking insights into tissue organization and function.

We employed SpaMI on eight simulated datasets to quantitatively demonstrate the model's robust ability to integrate spatial multi-omics data effectively. Further, we applied SpaMI to real datasets across various technological platforms and benchmarked it against competing methods. The results from these applications provide compelling evidence of SpaMI's superior capability in conducting high-precision and fine-grained analysis of tissue spatial domains. For instance, in the analysis of the mouse brain dataset, SpaMI delineated a more intricate layered structure within the cortical region compared to other methods. Similarly, SpaMI achieved a finer division of the cortex in the mouse thymus dataset, enhancing the interpretability of these complex tissue samples. The successful application of SpaMI across diverse modality combinations and multiple technology platforms also underscores its exceptional adaptability and compatibility, further validating its utility in spatial multi-omics research. Currently, the limited availability of spatial multi-omics datasets from abnormal or diseased biological samples has restricted the experimental validation of SpaMI in more complex and atypical biological environments. However, as spatial genomics technologies evolve, we anticipate the emergence of increasingly diverse datasets that will likely present new challenges and opportunities. We are confident that SpaMI will adeptly manage the complexities of these varied datasets.

Looking ahead, we aim to expand the scope of SpaMI beyond the integration of spatial multi-omics data from homologous cells to include cross-modality integration of spatial single-omics data obtained from different experimental batches or technological platforms. This expansion will enable us to fully leverage the wealth of data generated by contemporary spatial omics techniques. Furthermore, we plan to integrate high-resolution tissue images into our analyses, enhancing our model's ability to incorporate cellular morphology and achieve higher-resolution delineation of tissue structures.

## Methods

### Framework of SpaMI

SpaMI takes spatial multi-omics data matrices ($X_1 \in \mathbb{R}^{N \times d_1}$ and $X_2 \in \mathbb{R}^{N \times d_2}$) and spatial coordinates $Y \in \mathbb{R}^{N \times 2}$ as inputs, where $N$ denotes the number of spots, $d_1$ and $d_2$ represent the numbers of features such as genes, proteins or peaks. SpaMI contains following main modules: 1) data preprocessing; 2) construction of spatial neighbor graph; 3) omics-specific encoder; 4) attention layer for data integration; 5) omics-specific decoder. The specific content of each module will be introduced in detail below.

### Data preprocessing

We utilize the Scanpy [45] package to process the spatial transcriptome data by filtering out genes expressed in fewer than 10 cells, followed by logarithmic transformation and normalization. We then select highly variable genes for PCA, reducing the gene data to 50 dimensions for model input. For the spatial proteome, we apply centered log-ratio

normalization (CLR) [46] to process the protein count matrix. For the spatial epigenomic data, we filter out the peaks activated in less than 0.5% of cells. We then employ latent semantic indexing (LSI) [47] to reduce the dimensionality of this data to 50 dimensions. Specially, we filter out proteins expressed in fewer than 50 spots for the mouse thymus dataset, retaining 21 proteins. And the dimension of gene expression data is also reduced to 21 dimensions.

## Construction of spatial neighbor graph

We build spatial neighbor graph based on spatial coordinates and define the spatial graph as $G = (V, E)$, where $V = \{v_1, v_2, \ldots, v_N\}$ represents a set of $N$ spots, and $E$ represents the set of edges connecting the spots. For a given spot, we calculate its Euclidean distance to other spots based on the spatial coordinates and then select the $r$ closest spots as its neighbors, assuming that there is an edge between them, which can be formulated as follows:

$$D_i = \left\{ \sqrt{(v_{ix} - v_{jx})^2 + (v_{iy} - v_{jy})^2} \,\middle|\, j \in \{1, 2, \ldots, n\}, j \neq i \right\}$$

$$N_i = Sort(D_i, r)$$

where $D_i$ denotes the set of distances from spot $i$ to other spots, $v_{ix}$ and $v_{iy}$ denote the coordinates of the spot $i$, $N_i$ denotes the set of $r$ nearest neighbors of spot i. $Sort(\cdot)$ denotes the sort and select operation. We also generate a corrupted spatial neighbor graph $G' = (V', E')$ through data augmentation. Specially, we keep the topological structure of the spatial graph unchanged but randomly rearrange the rows of the initial feature matrix $X$ to create the corrupted spatial graph. $X'$ represents the corrupted feature matrix.

## Omics-specific encoder

In order to extract feature information unique to different omics, we use omics-specific encoder to encode each omics into a low-dimensional space. GCN learns potential representations by iteratively aggregating neighbor information and the node's own information, which can effectively capture the local neighborhood structure information of the node in the graph while retaining the global graph information. This property effectively combines modal expression data and spatial dependency information to capture the expression pattern of cells in the spatial structure. Before applying GCN for encoding, we set a dropout [48] layer to enhance the generalization ability of the simulation and prevent overfitting, which can be expressed as:

$$X_j = Dropout(X_j, p)$$

$$X_{j\_shuffle} = Dropout(X_{j\_shuffle}, p)$$

where $X_j$ and $X_{j\_shuffle}$ are the input feature matrix and perturbed feature matrix of $j$-th modality respectively, and $p$ is the dropout probability, which is set to 0.15 by default in our model. Then GCN is used for encoding, and the embedding of the $l$th layer in the encoder can be expressed as follows:

$$Z_1^{(l)} = \sigma(D^{-\frac{1}{2}} \widetilde{A} D^{\frac{1}{2}} Z_1^{(l-1)} W_1^{l-1} + b_1^{l-1})$$

$$Z_{1\_shuffle}^{(l)} = \sigma(D^{-\frac{1}{2}} \widetilde{A} D^{\frac{1}{2}} Z_{1\_shuffle}^{(l-1)} W_1^{l-1} + b_1^{l-1})$$

$$Z_2^{(l)} = \sigma(D^{-\frac{1}{2}}\widetilde{A}D^{\frac{1}{2}}Z_2^{(l-1)}W_2^{l-1} + b_2^{l-1})$$

$$Z_{2\_shuffle}^{(l)} = \sigma(D^{-\frac{1}{2}}\widetilde{A}D^{\frac{1}{2}}Z_{2\_shuffle}^{(l-1)}W_2^{l-1} + b_2^{l-1})$$

where $\widetilde{A}$ is the summation of the adjacent matrix $A$ with the unit matrix $I$, and $D$ is the degree matrix corresponding to $\widetilde{A}$. $W$ and $b$ denote the trainable weight matrix and the deviation vector, respectively, and $\sigma$ is the ReLU activation function. $Z_j^{(l)}$, $Z_{j\_shuffle}^{(l)}$ represent the representation and perturbed representation of all nodes in the $l$-th layer of the $j$-th modality, respectively. $Z_j^{(0)}$, $Z_{j\_shuffle}^{(0)}$ are set to $X_j$ and $X_{j\_shuffle}$, respectively.

### Attention layer for data integration

For each spot in a tissue sample, the expression patterns from different omics offer partial insights into the biological characteristics of the spot, containing both shared and unique elements. It is essential to preserve the commonalities across modalities while retaining the unique aspects of each one. Therefore, a simple concatenation of embeddings from different modalities is insufficient for integration. Naturally, we use the attention mechanism, through which model learns the importance of different modal data for parsing spots in tissue samples, and achieves adaptive aggregation of representations of different modalities. Specifically, we first concatenate the low-dimensional representations $Z_i$ of all points in different modalities, and then perform linear transformation and nonlinear activation processing on the concatenated data. The corresponding formulas are expressed as follows:

$$Z_{concat} = [Z_1 \parallel Z_2]$$

$$V = \tanh(W_v Z_{concat} + b_v)$$

where $\parallel$ denotes the concatenation operation, and $W_v$ and $b_v$ denote the trainable weight matrix and bias term, respectively. The attention coefficients are then computed based on the obtained transformed representation $V$ and normalized by softmax function to obtain the attention weights of different modalities for each sample spot:

$$U = W_u V + b_u$$

$$\alpha = softmax(U) = \frac{\exp(U)}{\sum \exp(U)}$$

where $W_u$ and $b_u$ are a set of trainable parameters, $\alpha$ denotes the different modal attention weights of all the sample spots, with higher values indicating higher contributions, and the sum of the attention weights of each sample spot is 1. Finally, the modal embeddings are weighted and summed according to the attention weights to obtain the fused output:

$$Z = Z_{concat} \times diag(\alpha)$$

where $diag(\alpha)$ converts $\alpha$ to a diagonal matrix. $Z$ is the low-dimensional representation of all spots after integrating the spatial multi-omics data, which can be used for downstream analysis such as visualization, clustering, spatial domain identification, and differential expression feature detection.

### The training procedure of SpaMI

SpaMI is trained jointly by three parts of loss functions, that are, contrastive learning loss, reconstruction loss and correspondence loss. The details of each part of loss are introduced as follows.

## Contrastive learning loss

To efficiently extract the features of each modality, we adopt a contrastive learning strategy to ensure that the spots in each modality can capture their local spatial context [49]. Specifically, we aggregate the modality-specific representations of the neighbors of spot $i$ and average the representations of the neighbors connected to spot $i$, then perform normalization to obtain the local representation $l_j^i$ of spot $i$ from the $j$ omics. Similarly, we can perform the neighborhood aggregation step on the spot representations of the perturbed graph to obtain the perturbed local representation $l_j^{i'}$. For spot $i$ in the graph, its representation $z_j^i$ forms a positive sample pair with the local representation $l_j^i$, while $z_j^i$ and $l_j^{i'}$ form a negative sample pair. Then we maximize the mutual information between positive sample pairs and minimize the mutual information between negative samples with the following binary cross entropy (BCE) loss:

$$\mathcal{L}_{CL} = -\frac{1}{2N}(\sum_{i=1}^{N} (\mathbb{E}_{(X,A)} \left[ \log \Phi \left( z_j^i, l_j^i \right) \right] + \mathbb{E}_{(X',A')}[\log(1 - \Phi \left( z_j^i, l_j^{i'} \right))]))$$

where $N$ is the total number of spots, $\Phi$ is a dual neural network discriminator that distinguishes between positive and negative pairs, and $\Phi \left( z_j^i, l_j^i \right)$ represents the probability score assigned to the corresponding positive pair. We also compute the same contrastive loss for the shuffled graph to enhance the balance of the model:

$$\mathcal{L}_{CL\_shuffle} = -\frac{1}{2N}(\sum_{i=1}^{N} (\mathbb{E}_{(X',A')} \left[ \log \Phi \left( z_j^{i'}, l_j^{i'} \right) \right] + \mathbb{E}_{(X,A)}[\log(1 - \Phi \left( z_j^{i'}, l_j^i \right))]))$$

Therefore, the final contrastive loss is:

$$\mathcal{L}_{contrast} = \gamma_1 \left( \mathcal{L}_{CL}^{omic\ 1} + \mathcal{L}_{CL\_shuffle}^{omic\ 1} \right) + \gamma_2 \left( \mathcal{L}_{CL}^{omic\ 2} + \mathcal{L}_{CL\_shuffle}^{omic\ 2} \right)$$

where $\gamma_1$ and $\gamma_2$ are hyperparameters that adjust the contrastive learning ability of the model.

## Reconstruction loss

We desire the model to have a better ability to preserve the expression patterns of individual modalities, and therefore designed modality-specific decoders to reconstruct the latent representation $Z$ of spots. The decoder uses a symmetric network structure of the GCN encoder to project $Z$ back into the original space of the different modalities. The modality-specific decoders are defined as follows:

$$Y_1^{(l)} = \sigma(D^{-\frac{1}{2}} \widetilde{A} D^{\frac{1}{2}} Y_1^{(l-1)} W_{r1}^{l-1} + b_{r1}^{l-1})$$

$$Y_2^{(l)} = \sigma(D^{-\frac{1}{2}} \widetilde{A} D^{\frac{1}{2}} Y_2^{(l-1)} W_{r2}^{l-1} + b_{r2}^{l-1})$$

where $\widetilde{A}$ is consistent with that in the encoder because the spatial graph topology remains unchanged, and $W_{ri}^{l-1}$ and $b_{ri}^{l-1}$ are trainable weight matrices and bias vectors in the decoder of different modalities. $Y_1^{(l)}$ and $Y_2^{(l)}$ represent the reconstructed profiles of the $l$-th layer of different omics. We train the model by minimizing the reconstruction loss, as follows:

$$\mathcal{L}_{rec} = \lambda_1(\frac{1}{N} \sum_{i=1}^{N} \left\| x_1^i - y_1^i \right\|_2^2) + \lambda_2(\frac{1}{N} \sum_{i=1}^{N} \left\| x_2^i - y_2^i \right\|_2^2)$$

where $\|\cdot\|_2$ represents the L2 norm, $N$ is the total number of spots, $x_1^i$ and $x_2^i$ are the features after PCA dimensionality reduction of the spot in different omics, $y_1^i$ and $y_2^i$ are the reconstructed spot features of different modality. $\lambda_1$ and $\lambda_2$ are weight factors used to adjust the reconstruction loss.

### Correspondence loss

Although the reconstruction loss can make the learned latent embedding $Z$ capture the information of different omics data, it cannot guarantee the similarity between $Z_1$ and $Z_2$. Considering that the modality-specific low-dimensional representations $Z_1$ and $Z_2$ obtained from different modalities both represent the same cells, we calculate the cosine coefficients between $Z_1$ and $Z_2$ to strengthen the similarity, which can be summarized as follows:

$$\mathcal{L}_{corr} = \mu(1 - \frac{Z_1 \cdot Z_2}{\|Z_1\| \, \|Z_2\|})$$

where $\mu$ is a hyperparameter used to adjust this loss.

Therefore, the overall loss function of SpaMI is defined as follows:

$$\mathcal{L}_{overall} = \mathcal{L}_{rec} + \mathcal{L}_{contrast} + \mathcal{L}_{corr}$$

### Implementation details and parameter selection

When creating the spatial graph, the number of neighbors per spot is set to 3 for the MISAR-seq and Stereo-CITE-seq datasets, 6 for the Spatial ATAC–RNA-seq dataset, and 3 for all simulated datasets. The weight factors $[\gamma_1, \gamma_2, \lambda_1, \lambda_2, \mu]$ in the loss function are set as follows: MISAR-seq dataset weight factors are $[1, 1, 10, 15, 3]$, Spatial ATAC–RNA-seq dataset weight factors are $[1, 1, 10, 15, 5]$, Stereo-CITE-seq dataset weight factors are $[1, 5, 10, 25, 1]$, and all simulated dataset weight factors are $[1, 1, 10, 10, 5]$, which is also the default parameter setting for the model. The learning rates are 0.01 and 0.001 in the MISAR-seq dataset and the Spatial ATAC–RNA-seq dataset respectively, and the training epochs are 800 and 400 respectively. The learning rate in the Stereo-CITE-seq dataset is 0.0001 and the training epochs is 600. In the simulated datasets, we set learning rate to 0.001 and the training epochs to 1,000. We compared the performance of SpaMI using above parameters and SpaMI with the default parameters on spatial ATAC-RNA-seq mouse brain data, which had reliable annotation based on the Allen brain atlases. We found that SpaMI still had better metric values than other methods (Fig G in S1 Text). We implemented all experiments on an Intel(R) 8269CY CPU and an NVIDIA RTX 3090 GPU. On the largest spatial ATAC-RNA-seq mouse brain data which contains 9215 spots, spaMI needs about three minutes and utilizes about 2200 MB of graphics memory. The detailed computational resource requirements of the SpaMI model on each dataset were shown in the Table B in S1 Text.

## Supporting information

**S1 Text. Supplementary materials.**
(DOC)

## Author contributions

**Formal analysis:** Congqiang Gao.

**Funding acquisition:** Lihua Zhang.

**Investigation:** Lihua Zhang.

**Methodology:** Congqiang Gao, Lihua Zhang.

**Project administration:** Lihua Zhang.

**Software:** Congqiang Gao.

**Supervision:** Lihua Zhang.

**Validation:** Congqiang Gao, Chenghui Yang, Lihua Zhang.

**Visualization:** Congqiang Gao, Chenghui Yang, Lihua Zhang.

**Writing – original draft:** Congqiang Gao, Lihua Zhang.

**Writing – review & editing:** Lihua Zhang.

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
