## [Decision Letter · Decision Letter 0]

20 Apr 2025

PCOMPBIOL-D-25-00362

A graph neural network-based spatial multi-omics data integration method for deciphering spatial domains

PLOS Computational Biology

Dear Dr. Zhang,

Thank you for submitting your manuscript to PLOS Computational Biology. After careful consideration, we feel that it has merit but does not fully meet PLOS Computational Biology's publication criteria as it currently stands. Therefore, we invite you to submit a revised version of the manuscript that addresses the points raised during the review process.

Please submit your revised manuscript within 60 days Jun 20 2025 11:59PM. If you will need more time than this to complete your revisions, please reply to this message or contact the journal office at ploscompbiol@plos.org. Please include the following items when submitting your revised manuscript:

We look forward to receiving your revised manuscript.

Kind regards,

Jean Fan

Academic Editor

PLOS Computational Biology

Jian Ma

Section Editor

PLOS Computational Biology

**Additional Editor Comments :**

While the editors do not view the concern about novelty raised by Reviewer #3 as a major issue, the authors are encouraged to clarify their contribution by differentiating more clearly from earlier work via descriptive explanation in addition to comparative evaluation/analysis.

**Journal Requirements:**

1) Please provide an Author Summary. This should appear in your manuscript between the Abstract (if applicable) and the Introduction, and should be 150-200 words long. The aim should be to make your findings accessible to a wide audience that includes both scientists and non-scientists. Sample summaries can be found on our website under Submission Guidelines:

Potential Copyright Issues:

i) Figure 1. Please confirm whether you drew the images / clip-art within the figure panels by hand. If you did not draw the images, please provide (a) a link to the source of the images or icons and their license / terms of use; or (b) written permission from the copyright holder to publish the images or icons under our CC BY 4.0 license. Alternatively, you may replace the images with open source alternatives. See these open source resources you may use to replace images / clip-art:

1) Please clarify all sources of financial support for your study. List the grants, grant numbers, and organizations that funded your study, including funding received from your institution. Please note that suppliers of material support, including research materials, should be recognized in the Acknowledgements section rather than in the Financial Disclosure

2) State the initials, alongside each funding source, of each author to receive each grant. For example: "This work was supported by the National Institutes of Health (####### to AM; ###### to CJ) and the National Science Foundation (###### to AM)."

3) State what role the funders took in the study. If the funders had no role in your study, please state: "The funders had no role in study design, data collection and analysis, decision to publish, or preparation of the manuscript."

4) If any authors received a salary from any of your funders, please state which authors and which funders.

6) Your current Financial Disclosure states, "The author(s) received no specific funding for this work."

However, your funding information on the submission form indicates receiving funds. Please ensure that the funders match between the Financial Disclosure field and the Funding Information tab in the online submission form. Both locations should list the same funders, grant numbers, and recipients in the same order.

**Reviewers' comments:**

Reviewer's Responses to Questions

Reviewer #1: The paper proposes SpaMI, a graph neural network-based method for integrating spatial multi-omics data to identify spatial domains. It combines contrastive learning and attention mechanisms to improve integration accuracy and denoising. However, several key issues need to be addressed:

1. It would be beneficial to briefly discuss the specific biological problems your study aims to address and their potential biological significance in the introduction. This would enhance the relevance and impact of your research.

2. The literature review section should be further enriched with a more detailed discussion of existing methods, particularly those directly related to the methods proposed in this paper.

3. As a computational model, the article does not provide detailed information on the computational resource requirements of the SpaMI model, including runtime and memory consumption.

4. For datasets without annotations, the authors used only Moran's I as the evaluation metric. Additional metrics should be employed to more comprehensively validate the performance of SpaMI.

5. The manuscript contains some textual errors, such as "import" on page 9. Please review the entire manuscript carefully.

6. Concerns remain about the experimental design. While the authors reported external clustering metrics (e.g., ARI, NMI) for simulated datasets, they only used internal metrics for real datasets. Given the significant differences between simulated and real data, the validation should be strengthened by including more annotated real datasets.

Reviewer #2: This manuscript introduces SpaMI, an innovative graph neural network (GNN)-based framework for integrating spatial multi-omics data. By leveraging contrastive learning, attention mechanisms, and joint optimization of reconstruction and correspondence losses, SpaMI effectively addresses the challenges of spatial heterogeneity and noise in multi-omics datasets. The authors validate SpaMI on both simulated and real-world datasets (e.g., mouse brain and thymus), demonstrating its superior performance in spatial domain identification, denoising, and cross-platform compatibility compared to existing methods. The work is timely, methodologically rigorous, and provides a valuable tool for spatial multi-omics research. The code and data availability further enhance its reproducibility and potential impact. I have the following suggestions.

1. Benchmarking on Public Datasets:

The authors mention multiple spatial multi-omics technologies (Refs. 7–16). Given the growing availability of public datasets, could SpaMI be tested on additional datasets (e.g., tumor tissues or human samples) to further validate its generalizability?

2. Estimation of Spatial Domain Number:

How is the number of spatial domains determined? Could automated methods (e.g., ASTER, scCCESS) be integrated to estimate domain numbers, similar to cell-type quantification in single-cell analysis?

3. Prior-Based Learning:

Public databases (e.g., 10x Visium, Allen Brain Atlas) host vast prior data. Can SpaMI leverage such resources to enhance integration via transfer learning or guided training (10.1101/gr.277891.123)?

4. Scalability via Metacell Strategies:

With increasing data scale, could metacell-based approaches (e.g., EpiCarousel) be adopted to improve computational efficiency while preserving spatial resolution?

5. Computational Efficiency:

The manuscript critiques spaMultiVAE’s complexity. Could the authors quantify SpaMI’s runtime, memory usage, and scalability on larger datasets?

6. Tool Packaging for Accessibility:

While the GitHub repository is valuable, packaging SpaMI into a Python library (e.g., via PyPi) with standardized APIs and tutorials would greatly enhance its usability and adoption.

Reviewer #3: This paper proposed a graph neural network-based model named SpaMI for spatial multi-omics data integration. SpaMI used contrastive learning strategy to extract features within each omics and introduced an attention mechanism to integrate omics-specific representations, which then are utilized for various downstream tasks. The model was validated on both simulated data and real spatial multi-omics datasets. Spatial multi-omics is cutting-edge technique, and while some methods have been developed for spatial multi-omics data integration, further development is still needed. Overall, this paper is well-written. However, the novelty is somewhat limited, as the SpaMI framework closely resembles that of a previous study (Long et al., 2024). There are some major concerns as below:

1. Insufficient novelty. SpaMI combines graph neural networks (GNN) with attention mechanism, which is quite similar to the approach in Long et al., 2024. Although contrastive learning is incorporated in SpaMI, the difference between the two methods is minimal

2. Weak results. The authors evaluated the performance of the proposed method using simulated and three real spatial multi-omics data. However, SpaMI appears comparable to the baseline, SpatialGlue, across most datasets. For example,

Visually, the spatial distribution and UMAP results for the simulation show that SpaMI and SpatialGlue are comparable. Both methods accurately identify the four factors and background.

In the mouse thymus results, it is difficult to discern any significant advantage of SpaMI over SpatialGlue, as both methods identify main tissue regions with high accuracy.

3. Inaccurate or unclear descriptions:

The author claim that single cell multi-omics integration methods might not be applicable for spatial multi-omics data as spatial information is not considered. However, these methods can still be applied to spatial multi-omics data, as one can choose not to use spatial information.

The author state, “However, the model adopts single-layer GCN as encoder, making the network structure too simple to effectively capture complex structures”. Are there any supporting evidence or results for this claim?

4. Missing labels. Some figures lack labels, such as Fig.3 C, E, Fig. 4 C, E, Fig. 6 C, F. The authors provide labels for SpaMI, but not for other methods.

5. How about the robustness of the proposed method? Most parameters vary across different datasets, which might weaken the generalizability of the proposed method when applied to new datasets.

6. Some typos can be found in the manuscript, such as “is decrease”.

**Have the authors made all data and (if applicable) computational code underlying the findings in their manuscript fully available?**

Reviewer #1: None

Reviewer #2: None

Reviewer #3: Yes

PLOS authors have the option to publish the peer review history of their article (what does this mean? ). If published, this will include your full peer review and any attached files.

**Do you want your identity to be public for this peer review?** For information about this choice, including consent withdrawal, please see our Privacy Policy .

Reviewer #1: No

Reviewer #2: No

Reviewer #3: No

**Figure resubmission:**
---

## [Decision Letter · Decision Letter 1]

15 Aug 2025

PCOMPBIOL-D-25-00362R1

A graph neural network-based spatial multi-omics data integration method for deciphering spatial domains

PLOS Computational Biology

Dear Dr. Zhang,

Thank you for submitting your manuscript to PLOS Computational Biology. After careful consideration, we feel that it has merit but does not fully meet PLOS Computational Biology's publication criteria as it currently stands. Therefore, we invite you to submit a revised version of the manuscript that addresses the points raised during the review process.

Please submit your revised manuscript within 30 days Oct 15 2025 11:59PM. If you will need more time than this to complete your revisions, please reply to this message or contact the journal office at ploscompbiol@plos.org. Please include the following items when submitting your revised manuscript:

We look forward to receiving your revised manuscript.

Kind regards,

Shaun Mahony

Section Editor

PLOS Computational Biology

Shaun Mahony

Section Editor

PLOS Computational Biology

**Additional Editor Comments:**

Thank you for sending your revised manuscript back to PLOS Computational Biology. As you will see, one of the reviewers has remaining concerns. These issues will need to be addressed before we can consider publication.

**Journal Requirements:**

1) Please provide an Author Summary. This should appear in your manuscript between the Abstract (if applicable) and the Introduction, and should be 150-200 words long. The aim should be to make your findings accessible to a wide audience that includes both scientists and non-scientists. Sample summaries can be found on our website under Submission Guidelines:

2) We have noticed that you have uploaded Supporting Information files, but you have not included a list of legends. Please add a full list of legends for your Supporting Information files after the references list.

3) Please ensure that the funders and grant numbers match between the Financial Disclosure field and the Funding Information tab in your submission form. Note that the funders must be provided in the same order in both places as well.

State what role the funders took in the study. If the funders had no role in your study, please state: "The funders had no role in study design, data collection and analysis, decision to publish, or preparation of the manuscript.".

**Reviewers' comments:**

Reviewer's Responses to Questions

**Comments to the Authors:**

Reviewer #2: My concerns have been addressed.

Reviewer #3: The authors have carefully addressed most of my comments. However, the manuscript still requires further improvement before it can be considered for publication.

1. While the authors have quantitatively demonstrated the contributions of each components using simulated data, these results are insufficient. First, no visualization results are provided. Quantitative evaluation alone may not objectively reflect model performance. Visual evaluation should be also included to support the findings. Second, the ablation study is limited to simulated data, which is overly simplistic and may not accurately capture the complexity of real-world scenarios. Therefore, the inclusion of real-world datasets in the ablation study is necessary to more rigorously validate the model’s effectiveness.

2. In their response, the authors noted that prior information such as annotations was used in the extended model. However, this detail appears to be missing from the manuscript. If such supervision was indeed employed, the comparisons with baseline methods may be biased, as many of the baselines are unsupervised. This discrepancy should be clarified to ensure a fair evaluation.

3. The figures should be improved. For example, the labels are provided only for the proposed methods, such as Fig. 2A, Fig. 3C. The labels for other methods should be also provided.

**Have the authors made all data and (if applicable) computational code underlying the findings in their manuscript fully available?**

Reviewer #2: None

Reviewer #3: Yes

PLOS authors have the option to publish the peer review history of their article (what does this mean? ). If published, this will include your full peer review and any attached files.

**Do you want your identity to be public for this peer review?** For information about this choice, including consent withdrawal, please see our Privacy Policy .

Reviewer #2: **Yes: ** Shengquan Chen

Reviewer #3: No

**Figure resubmission:**
---

## [Decision Letter · Decision Letter 2]

22 Sep 2025

Dear Dr. Zhang,

We are pleased to inform you that your manuscript 'A graph neural network-based spatial multi-omics data integration method for deciphering spatial domains' has been provisionally accepted for publication in PLOS Computational Biology.

Best regards,

Shaun Mahony

Section Editor

PLOS Computational Biology

Reviewer #2:

Reviewer #3:

Reviewer's Responses to Questions

**Comments to the Authors:**

Reviewer #2: I have no further concerns.

Reviewer #3: The authors have carefully addressed the comments. I recommend the publication of this paper.

**Have the authors made all data and (if applicable) computational code underlying the findings in their manuscript fully available?**

Reviewer #2: None

Reviewer #3: Yes

PLOS authors have the option to publish the peer review history of their article (what does this mean? ). If published, this will include your full peer review and any attached files.

**Do you want your identity to be public for this peer review?** For information about this choice, including consent withdrawal, please see our Privacy Policy .

Reviewer #2: No

Reviewer #3: No

---

## [Editor Report · Acceptance letter]

PCOMPBIOL-D-25-00362R2

A graph neural network-based spatial multi-omics data integration method for deciphering spatial domains

Dear Dr Zhang,

I am pleased to inform you that your manuscript has been formally accepted for publication in PLOS Computational Biology. Your manuscript is now with our production department and you will be notified of the publication date in due course.

With kind regards,

Zsofia Freund
